# Integrated Model Explanations by Independent and Collaborative Feature Influence via Linear-Nonlinear Perspectives

## Abstract

In machine learning, model-agnostic explanation methods try to give explanation to model prediction by assessing the importance of input features. While linear simplification methods guarantee good properties, they have to include nonlinear feature interactions into linear coefficients. On the other hand, feature influence analysis methods examine feature relevance, but do not consistently preserve the desirable properties for robust explanations. Our approach seeks to inherit properties from linear simplification methods while systematically capturing feature interactions. To achieve this, we consider the explained model from two aspects: the linear aspect, which focuses on the independent influence of features to model predictions, and the nonlinear aspect, which concentrates on modeling feature interactions and their collaborative impact on model predictions. In practice, our method initially investigates both the linear and nonlinear aspects of the model being explained. It then extracts the independent and collaborative importance of features on model predictions and consistently combines them to ensure that the resulting feature importance preserves the desirable properties for robust explanations. Consequently, our Linear-Nonlinear Explanation (LNE) method provides a comprehensive understanding on how features influence model predictions. To validate its effectiveness, experiments demonstrate that linear, nonlinear, and the combined feature importance all offer valuable insights for explaining model predictions. We also compare the performance of LNE with other methods on explaining well-trained classifiers, and find our explanations align more closely with human intuitions. Additionally, user study shows our method can hint humans with potential biases in classifiers.

## 1 Introduction

Due to their black-box nature, machine learning models can easily contain biases and errors that often go unnoticed by humans. This necessitates the development of explanation methods to peek inside these opaque models and understand their inner workings. Among numerous approaches, model-agnostic explanation methods don't rely on specific structure or type of the model being explained. This makes them particularly advantageous and straightforward to implement (Arrieta et al., 2020). However, due to the lack of detailed information, conventional model-agnostic explanation methods typically involve simplifying the explained model using a highly interpretable proxy, or analyzing feature influence to the model predictions. Do these methods faithfully adhere to the original model and provide accurate explanations?

There have been some methods aimed at explaining models through simplification. One such method is LIME (Ribeiro et al., 2016), which locally approximates the explained model using a linear model centered around the input instance. However, (Zhang et al., 2019) claims LIME's interpretations introduce uncertainty for different data types due to its sampling procedure. Subsequently, SHAP (Lundberg & Lee, 2017) has highlighted that the heuristic selection of parameters in LIME struggles to provide guaranteed properties. SHAP addresses this by unifying similar approaches to propose additive feature attribution methods employing the Shapley value (Shapley, 1953) from cooperative game theory. It guarantees three desirable properties, i.e. (1) local accuracy: simplified model matches output of original model; (2) missingness: features missing in input

have no impact to output; (3) consistency: feature contribution increases caused by original model changes should reflect on simplified model. Computing Shapley values requires checking all subsets of features within the input instance, thus is NP-hard. Therefore, SHAP makes an assumption of feature independence, and approximate Shapley values using Shapley kernel weighted linear regression. Nevertheless, (Aas et al., 2021) points out this independence assumption results in misleading explanations when features are correlated. Furthermore, (Alvarez-Melis & Jaakkola, 2018) claims when dealing with complex models like neural network classifiers, both LIME and SHAP explanations can exhibit significant variation within very small neighborhoods and may be inconsistent with each other. There have other been attempts. (Tan et al., 2018) introduces a distillation way to extract transparent version of the original model. In specific, they simplify original model to iGAM (Lou et al., 2013). It adds pair-wise feature interaction terms to addictive model, thus describes nonlinear aspect of the explained model. However, feature interactions could be very complex, whether pair-wise interactions cover higher order interactions needs further exploration. To give thorough study on nonlinear aspect caused by feature interactions, some works have abandoned the simplification way, and turn into analyzing feature influence to model prediction (Cortez & Embrechts, 2013; Datta et al., 2016). They apply sensitivity analysis, aggregating marginal influence, or cooperative game theory to reveal the complex nonlinear relationships between features. But these methods typically require extra information such as a baseline vector, distribution of the inputs, or datasets on which the original model is trained. It makes them less easy to implement compared with local simplification methods. Additionally, they don't satisfy desirable properties all the time.

To sum up, linear simplification method like SHAP (Lundberg & Lee, 2017) considers cooperative effects of features, and guarantees local accuracy, missingness and consistency of the feature attribution. It tries to include nonlinear feature interactions into linear coefficients. As previous works have pointed out, this design contains much uncertainty. Nonlinear simplification methods like Distillation (Tan et al., 2018) or iGAM (Lou et al., 2013) also preserve local accuracy. They try to inadequately include nonlinear part in the simplified model, by only considering pair-wise interactions. Feature influence analyzing methods give detailed investigation on how features interact, but they don't always satisfy desirable properties. Therefore, we seek for a connection point for the previous works. We'd like to propose an method that inherits good properties from linear simplification methods, and in the meantime systematically describes feature interactions as well. In our design, the explained model is considered as two aspects: linear aspect, in which features independently contribute to model prediction, thus handy to simplify it using additive model. Nonlinear aspect, in which features collaboratively contribute to output, and feature influence analyzing method is applied to check feature's importance. Importantly, the two aspects information must be combined together consistently. We achieve this by LNE (Linear-Nonlinear Explanations): (1) simultaneously investigate linear and nonlinear aspect of the explained model, (2) correspondingly design indicators to extract feature importance from linear and nonlinear aspects, (3) consistently combine the two aspects to get a comprehensive understanding of feature importance, so that they still preserve desirable properties. See Figure (1) for an overview of the LNE method.

Specifically, the linear aspect focuses on features' independent influence to model prediction. We do this by approximating the explained model with an additive model. Since we only concentrate on independent feature importance in the linear aspect, by assuming feature independence, we can linearly decompose the explained model, and take the coefficients as linear indicator of feature importance. Except for independent influence, features also collaboratively contribute to model prediction. Thus, the nonlinear aspect focuses on non-linearity caused by feature interactions. To construct the nonlinear indicator showing features' collaborative importance, we follow the classic way to aggregate marginal influence for different subsets of features like in (Datta et al., 2016). After obtaining independent and collaborative importance of feature, we combine them to get comprehensive information. Because they stand for distinct meanings, we have to unify their scales but keep the sign unchanged for each importance value. Therefore, we use a hyperbolic tangent function to unify their scales to $[-1, 1]$ and maintain the original signs. Since we treat both aspects equally important, we average the rescaled values to combine them. It proves that the combined linear-nonlinear feature importance preserves desirable properties of missingness and a conditional consistency. Experiments show dynamically that the linear and nonlinear indicators both provide insights to explain model predictions, and the combined indicator provides a comprehensive understanding. We also compare LNE's performance with other methods, and find our explanation for well-trained classifiers are more consistent with human intuitions. Additionally, user study shows our method can hint humans with potentially biased classifiers.

Our main contributions states as follows:

- Instead of simplifying the explained model to a specific form, we investigate linear and nonlinear aspects of the model simultaneously. They stands for features contributing both independently and collaboratively to model prediction.
- We construct meaningful indicators for both aspects. The linear indicator measures feature's independent importance, while the nonlinear indicator represents feature's collaborative importance.
- The two aspects of importance combine consistently to provide comprehensive understanding of feature's influence. And they still preserve desirable properties. Thereby, we give explanation to any classifiers by the combined linear-nonlinear feature importance values.

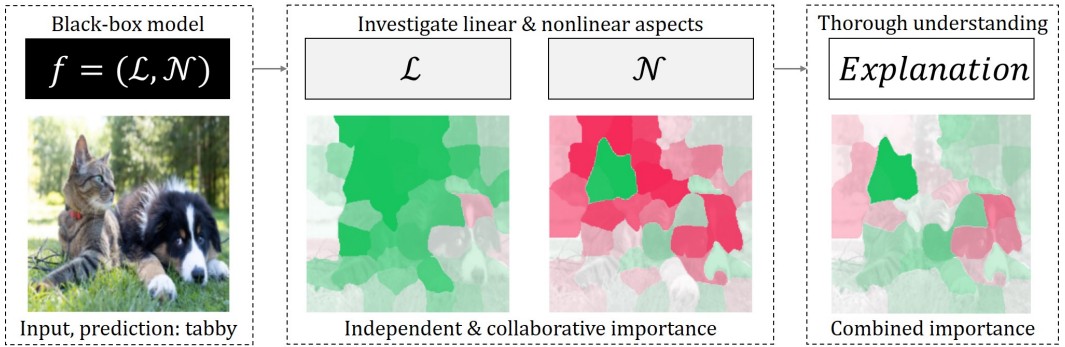

Figure 1: Overview

## 2 HOW LOCAL EXPLANATION METHOD WORKS

Before diving into the proposed LNE method, we'd like to go through some basic concepts on local explanation method. It will help understanding the mechanism of our method.

### 2.1 MODEL & INSTANCE TO BE EXPLAINED

Local explanation method tries to explain a model $f$ locally at input $\boldsymbol{x}$, figure out how it leads to prediction $f(\boldsymbol{x})$. Specifically, $f$ is a classifier in most cases, since classification task is the fundamental task in machine learning. A typical classifier is a vector function $\boldsymbol{f}$, which predicts probabilities of input $\boldsymbol{x}$ belonging to certain class over all $n$ labels:

$$\boldsymbol{f}(\boldsymbol{x}) = (f_1(\boldsymbol{x}), f_2(\boldsymbol{x}), \cdots, f_i(\boldsymbol{x}), \cdots, f_n(\boldsymbol{x})), \ f_i(\boldsymbol{x}) \in \mathbb{R}$$

For example, a classifier on the MINST dataset (Deng, 2012) predicts 10 real numbers based on input image. Each one represents the probability of input being classified to number $0 - 9$. One certain label function $f_i$ is referred to as the explained model $f$. Correspondingly, $\boldsymbol{x}$ is called the explained instance. We talk about model-agnostic methods in this paper, which means specific structures or parameters of $f$ are unknown. With so little information, one has to work with interpretable features within $\boldsymbol{x}$ to give explanation.

### 2.2 INTERPRETABLE FEATURES

Interpretable features are attributions in input $\boldsymbol{x}$ that humans can understand. Such features are various for different datatype: for table data, the column features are interpretable. For text, its interpretable features are naturally words. For image, they can be segmented image patches. Assume the explained instance $\boldsymbol{x}$ contains $m$ interpretable features, it can be represented as set:

$$\boldsymbol{x} = \{x_1, x_2, \cdots, x_i, \cdots, x_m\} \tag{1}$$

With these features, explanation method will check each feature's influence to model prediction. To do this, the so-called **perturbed instance** $z$ will be sampled by blocking some features in $x$. A binary vector $z'$ will indicate the presence of features:

$$z \subseteq x, \ z' = (z_1, z_2, \cdots, z_i, \cdots, z_m), \ z_i \in \{0, 1\} \tag{2}$$

in which $z_i = 0$ indicates feature $x_i$ of $x$ is blocked. For instance, if $x$ is an image, then the $i$-th image patch will be replaced by meaningless background in $z$. If all elements of $z'$ equal one, $z = x$. We also call $z$ as **subset** of $x$. Explanation methods utilize $z$ to check importance of each feature $x_k$ in different ways, as shown below.

### 2.3 Explain feature importance

Take the addictive feature attribution method (Lundberg & Lee, 2017) as example. It utilizes linear model $g$ to simplify $f$ centered around $x$. It tries to minimize discrepency between $f, g$ on sampled perturbed instances $z$ and ensure local accuracy, i.e. $f(z) = g(z')$ when $z = x$.

$$g(z') = \phi_0 + \sum_{i=1}^{m} \phi_i z_i, \ \phi_i \in \mathbb{R}, \ z_i \in \{0, 1\} \tag{3}$$

Specifically, SHAP (Lundberg & Lee, 2017) proves only when coefficients of $g$ are Shapley values (Shapley, 1953), $g$ will satisfy desirable properties of local accuracy, missingness, and consistency. $\phi_i$ is formulated as:

$$\phi_i = \sum_{z \subseteq x \setminus [x_k]} \frac{(|z| + 1)!(m - |z|)!}{m!} [f(z + [x_k]) - f(z)] \tag{4}$$

where $|z|$ stand for number of features within $z$. $[x_k]$ is the perturbed instance consisting of feature $x_k$ alone. $z + [x_k], x \setminus [x_k]$ stand for the perturbed instances adding, dropping $x_k$ respectively. Particularly, $\mathbf{0}$ stands for no features at all, and $f(\mathbf{0}) = 0$. The term of $[f(z + [x_k]) - f(z)]$ is the marginal influence. It is applied to analyze feature influence in method like (Datta et al., 2016). Consequently, $\{\phi_i\}$ explains $f$ at $x$ by showing each feature's importance value.

## 3 Linear-Nonlinear Explanation method

In this section, we propose our Linear-Nonlinear Explanation (LNE) method to explain any classifiers. We consider the explained model as two aspects, $f = (\mathcal{L}, \mathcal{N})$. $\mathcal{L}$ is the linear aspect, which models features' independent importance to $f$. $\mathcal{N}$ is the nonlinear part, which simulates features' collaborative importance to $f$. In our design, to get a comprehensive understanding of feature influence to model prediction, we will go through three steps: (1) investigate linear and nonlinear aspect simultaneously. (2) construct corresponding indicators to extract feature importance from the two aspects. (3) combine the two indicators in a consistent way to get linear-nonlinear feature importance value, so that they still preserve desirable properties. An overview of the proposed LNE method can be seen in Figure 1. We start from the linear aspect.

### 3.1 Linear aspect

In linear aspect $\mathcal{L}$ of the explained model, we focuses on feature's independent influence to model prediction. Thus, we will first define feature independence. As we will see, by assuming feature independence, we are able to give an linear decomposition of the explained model. It approximates $f$ at $x$ with local accuracy. This linear decomposition also reveals independent importance of features.

#### 3.1.1 feature independence

As a simple intuition, feature independence refers to a constant marginal influence by adding feature $x_k$ to any subsets $z \subset x$, where $x_k \notin z$. Conventional marginal influence is defined by $f(z + [x_k]) - f(z)$, where the group feature influence is measured by $f(z)$. However, we argue that $f(z)$ is not enough to judge $z$'s influence from a class label information coverage prospective. Consider $\bar{z} = x \setminus z$, i.e. complement of $z$ according to $x$. Let $z^*$ already contains all features that contribute highly to $f$'s prediction, then features in $\overline{z^*}$ contribute little, $f(\overline{z^*}) = f(x \setminus z^*) \to 0$. Conversely,

if $z$ and $\bar{z}$ both includes some deterministic features, then $f(\bar{z})$ can be as large as $f(z)$. Because the numbers of important features are different, $z$ carries less information than $z^*$, and is not as influential as $z^*$. Therefore, we define the influence for group feature by checking the gap between it and its complement. See the following definition.

**Definition 1. (Influence of group feature)** Let $z = \{x_{i_1}, x_{i_2}, \cdots, x_{i_l}, \cdots, x_{i_s}\}, x_{i_l} \in x, 1 \le i_l \le m$. $\bar{z}$ is the complement of $z$ according to $x$. Then, the group influence of $z$ is defined as:

$$F(z) := f(z) - f(\bar{z}) \tag{5}$$

It is easy to check by this influence definition, $F(\bar{z}) = -F(z)$. Complement of $z$ has precisely the opposite influence as $z$ does. It considers $z$'s influence to $f$ from both side of the coin, thus it carries more information than conventional influence indicator $f(z)$. We are now prepared to define the feature independence using (5).

**Definition 2. (Feature independence)** For any $x_k \in x$, and $z \subseteq y \subset x$, in which $x_k \notin z$, feature independence indicates constant marginal influence by adding $x_k$:

$$F(z + [x_k]) - F(z) = F(y + [x_k]) - F(y) \tag{6}$$

### 3.1.2 LINEAR DECOMPOSITION OF THE EXPLAINED MODEL

Following the design of our method, we seek to approximate the explained model $f$ by an additive model with feature independence assumption (Definition 2). The following theorem shows how to obtain the additive model by linearly decomposing $f$.

**Theorem 1. (Linear decomposition of explained model)** With feature independence, $f$ can be linearly decomposed as $f(z)|_{z=x} = l(z') = \sum_{i=1}^{m} \phi_i z_i$, where $\phi_i = \dfrac{F([x_i]) + f(x)}{2}$.

*Proof.* With feature independence, for any $x_k \notin z \subseteq y \subset x$, it holds $F(y + [x_k]) - F(y) = F(z + [x_k]) - F(z)$. We can decompose $F$ by:

$$F(x) - F(x\backslash[x_1]) = F([x_1]) - F(\emptyset)$$
$$F(x\backslash[x_1]) - F(x\backslash[x_1 + x_2]) = F([x_2]) - F(\emptyset)$$
$$\cdots$$
$$F(x\backslash[x_1 + \cdots + x_k]) - F(x\backslash[x_1 + \cdots + x_{k+1}]) = F([x_{k+1}]) - F(\emptyset)$$
$$\cdots$$
$$F([x_{m-1} + x_m]) - F([x_m]) = F([x_{m-1}]) - F(\emptyset)$$

Therefore, $F(x) = \sum_{i=1}^{m} F([x_i]) - (m-1)F(\emptyset) = F(\emptyset) + \sum_{i=1}^{m}(F([x_i]) - F(\emptyset))$. Consequently, $f(z)|_{z=x} = F(x) = \phi_0 + \sum_{i=1}^{m} \phi_i$, in which $\phi_0 = F(\emptyset) = -f(x), \phi_i = F([x_i]) - F(\emptyset) = F([x_i]) + f(x)$. Move $\phi_0$ to left side, and get $f(z)|_{z=x} = l(z') = \sum_{i=1}^{m} \dfrac{F([x_i]) + f(x)}{2} z_i$ $\qquad \square$

It is important to note this decomposition only holds when $z = x$. It is not true when $z \neq x$, because $f(z) = F(z) + f(\bar{z})$, in which $F(z)$ can be decomposed while $f(\bar{z})$ cannot. Thus, the additive model $l(z')$ approximates $f$ locally at $x$ with local accuracy. Since $l(z')$ is directly derived from feature independence assumption, its coefficients represent feature's independent importance to model prediction. Therefore, we obtain the linear indicator.

### 3.1.3 LINEAR INDICATOR

Linear indicator is responsible for measuring independent contribution of features. This indicator directly comes from linear decomposition coefficients in Theorem 1, namely $\mathcal{F}([x_k]) + f(x) = f([x_k]) - f(x\backslash[x_k]) + f(x)$. On one hand, it stands for feature $x_k$'s importance in the linear aspect of $f$. On the other hand, this term itself is meaningful. $f([x_k])$ directly shows $x_k$'s importance by its value; higher value means more importance. $f(x) - f(x\backslash[x_k])$ shows the marginal loss when $x$ drops this feature. $f([x_k]) - f(x\backslash[x_k]) = F([x_k])$ is the influence of $x_k$ by Definition 1. Therefore, this term is meaningful, and it describes feature's independent importance from the linear aspect. Consequently, we have the linear indicator of feature $x_k$ as $L(x_k)$:

$$L(x_k) = \frac{f([x_k]) + f(x) - f(x\backslash[x_k])}{2} \tag{7}$$

## 3.2 NONLINEAR ASPECT

After analyzing the linear aspect, we turn to the nonlinear aspect $\mathcal{N}$, which concentrates on feature's collaborative importance caused by feature interactions. We first model the feature interactions. Then, we follow a classic way to aggregate marginal influence for different subsets of features. Finally, we construct the nonlinear indicator measuring collaborative importance of features.

### 3.2.1 FEATURE INTERACTIONS

Unlike linear aspect, it is impractical to approximate nonlinear aspect of $f$ with any specific forms. However, it is still possible to model the process by dividing them according to the number of features involving interactions:

$$\mathcal{N}(\boldsymbol{x}) = \sum_{i<j} \mathcal{I}(x_i, x_j) + \sum_{i<j<k} \mathcal{I}(x_i, x_j, x_k) + \cdots + \mathcal{I}(x_1, x_2, \ldots, x_m) \tag{8}$$

in which $\mathcal{I}(\cdot)$ represents the interaction of features. For feature $x_k$, its interactions with others are:

$$\mathcal{N}(\boldsymbol{x}; x_k) = \sum_{i_1 \neq k} \mathcal{I}(x_k, x_{i_1}) + \sum_{i_1, i_2 \neq k} \mathcal{I}(x_k, x_{i_1}, x_{i_2}) + \cdots + \mathcal{I}(x_k, x_{i_1}, \ldots, x_{i_{m-1}})$$

$$= \frac{1}{2}\Big(\sum_{i_1 \neq k} \mathcal{I}(x_k, x_{i_1}) + \mathcal{I}(x_k, x_{i_2}, \cdots, x_{i_{m-1}})\Big) + \frac{1}{2}\Big(\sum_{i_1, i_2 \neq k} (\mathcal{I}(x_k, x_{i_1}, x_{i_2}) + \mathcal{I}(x_k, x_{i_3}, \cdots, x_{i_{m-1}}))\Big)$$

$$+ \cdots + \frac{1}{2}\Big(\sum_{i_1, i_2, \cdots i_{m-2} \neq k} \mathcal{I}(x_k, x_{i_1}, \cdots, x_{i_{m-2}}) + \mathcal{I}(x_k, x_{i_{m-1}})\Big) + \mathcal{I}(x_k, x_{i_1}, \ldots, x_{i_{m-1}})$$

Detailed form of $\mathcal{I}$ could be very complex to investigate. However, we only concern certain feature $x_k$'s importance within $\mathcal{I}$. It can be expressed by derivative of $\mathcal{I}$ according to $x_k$, which means the variation of model prediction with & without $x_k$. Specifically, such derivative can be formulated by marginal influence adding $x_k$:

$$\frac{\partial}{\partial x_k} \mathcal{I}(x_k, x_{i_1}, \cdots, x_{i_s}) := (f(\boldsymbol{z} + [x_k]) - f(\boldsymbol{z}))P(\boldsymbol{z}),\ x_{i_h} \in \boldsymbol{z}, h = 1, 2, \cdots, s \tag{9}$$

in which $P(\boldsymbol{z})$ represents the probability of $\boldsymbol{z}$ as subset of $\boldsymbol{x}$. We follow the Banzhaf index (Banzhaf III, 1964) to consider $P(\boldsymbol{z}) = 2^{-(m-1)}$, which means all subsets of $\boldsymbol{x}\backslash[x_k]$ equally involve and uniformly aggregate. Therefore, we formulate collaborative importance of $x_k$ as $\mathcal{N}(x_k)$:

$$\mathcal{N}(x_k) := \frac{\partial}{\partial x_k} \mathcal{N}(\boldsymbol{x}; x_k) = \sum_{1 \leq |\boldsymbol{z}| \leq m-1} N(x_k, \boldsymbol{z})P(\boldsymbol{z}) = \sum_{d=1}^{m-1} \sum_{|\boldsymbol{z}|=d} N(x_k, \boldsymbol{z})P(\boldsymbol{z}) \tag{10}$$

where $N(x_k, \boldsymbol{z}) = \frac{1}{2}[f(\boldsymbol{z} + [x_k]) - f(\boldsymbol{z}) + f(\bar{\boldsymbol{z}}) - f(\bar{\boldsymbol{z}}\backslash[x_k])]$.

### 3.2.2 NONLINEAR INDICATOR

Precisely computing (10) consumes exponential time. The complexity originates from choosing subset $\boldsymbol{z}$ from $\boldsymbol{x}\backslash[x_k]$. In practice, to construct an indicator to measure collaborative importance of $x_k$, we just draw approximate information from (10). Specifically, instead of checking numerous $N(x_k, \boldsymbol{z})$ with $|\boldsymbol{z}| = d$, we use a meaningful $\boldsymbol{z}_d$ to delegate them, and formulate the approximation:

$$N(x_k) = \sum_{d=1}^{m-1} \sum_{|\boldsymbol{z}|=d} N(x_k, \boldsymbol{z}_d)P(\boldsymbol{z}) = \sum_{d=1}^{m-1} N(x_k, \boldsymbol{z}_d) \sum_{|\boldsymbol{z}|=d} P(\boldsymbol{z}) \tag{11}$$

$$\approx \sum_{d=1}^{m-1} N(x_k, \boldsymbol{z}_d) \frac{1}{\sqrt{\pi(m-1)/2}} \exp\{-\frac{(d - ((m-1)/2))^2}{(m-1)/2}\} \tag{12}$$

where $\approx$ in (12) is the Gaussian approximation of binomial distribution when $m$ is large, within $\mathcal{O}(\frac{1}{\sqrt{n}})$ error. Each $\boldsymbol{z}_d$ is obtained dynamically. Until $s$ step, the cumulative value of $N^s(x_k) = \sum_{d=1}^{s} N(x_k, \boldsymbol{z}_d) \frac{1}{\sqrt{\pi(m-1)/2}} \exp\{-\frac{(d-((m-1)/2))^2}{(m-1)/2}\}$. $\boldsymbol{z}_{s+1}$ is then obtained by adding the feature $x_i$ with the maximum $N^s(x_i)$ value to $\boldsymbol{z}_s$. In case $x_k \in \boldsymbol{z}_d$, the term $N(x_k, \boldsymbol{z}_d) = \frac{1}{2}[f(\boldsymbol{z}) -$

$f(\boldsymbol{z}\backslash[x_k]) + f(\bar{\boldsymbol{z}} + [x_k]) - f(\bar{\boldsymbol{z}})]$. The series of $\boldsymbol{z}_1, \boldsymbol{z}_2, \cdots, \boldsymbol{z}_{m-1}$ stands for greedy pick of features with the highest collaborative importance. This makes features in each $\boldsymbol{z}_d$ interact fiercely, so that marginal influence performed on it shows significant characteristics. We called the series as the **sampling trace** for computing collaborative importance. Consequently, $N(x_k)$ is called nonlinear indicator of $x_k$.

### 3.3 COMBINE LINEAR-NONLINEAR INDICATORS

The linear and nonlinear indicators $L(x_k), N(x_k)$ have distinct meanings and different scales. They both show one aspect of the explained model, and only together they offer comprehensive understanding of feature importance. To combine them, we have to re-scale the two and keep each values' sign unchanged. Hyperbolic tangent function maps input to $[-1, 1]$ without changing the sign, thus perfectly meet our need. Since we consider linear and nonlinear aspect equally important, we average the rescaled values to combine them.

$$C(x_k) = \frac{1}{2}(tanh(L(x_k)) + tanh(N(x_k))), \ tanh(x) = \frac{e^x - e^{-x}}{e^x + e^{-x}} \tag{13}$$

$C(x_k)$ is the linear-nonlinear feature importance of $x_k$. We will show in the next theorem, that it preserves the desirable properties of missingness and a conditional consistency.

**Theorem 2.** The combined linear-nonlinear feature importance $C(x_k)$ satisfies properties: (1) missingness, features missing in the original input $\boldsymbol{x}$ have no impact to model output. $z_i = 0 \Rightarrow C(x_i) = 0$. (2) conditional consistency, for any two models $f, f'$, if $f'(\boldsymbol{z}+[x_k]) - f'(\boldsymbol{z}) \geq f(\boldsymbol{z}+[x_k]) - f(\boldsymbol{z})$ holds for all $\boldsymbol{z} \subseteq \boldsymbol{x}$, then with fixed sampling trace $\boldsymbol{z}_1, \boldsymbol{z}_2, \cdots, \boldsymbol{z}_{m-1}, C'(x_k) \geq C(x_k)$.

*Proof.* Missingness is obvious. If there are missing features in $\boldsymbol{x}$, they won't involve into computation of the linear and nonlinear indicator. Let $\boldsymbol{z} = \emptyset$ and $\boldsymbol{z} = \boldsymbol{x}\backslash[x_k]$, then $L'(x_k) \geq L(x_k)$. With fixed sampling trace, $N'(x_k, \boldsymbol{z}_d) \geq N(x_k, \boldsymbol{z}_d)$ is always true, thus $N'(x_k) \geq N(x_k)$. Since $tanh(\cdot)$ is monotonically increasing, $C'(x_k) \geq C(x_k)$. $\square$

Therefore, our LNE method still preserves some good properties. We propose Algorithm 1 for our method. The time complexity is $\mathcal{O}(m^2)$, space complexity is $\mathcal{O}(m)$. Therefore, it can be implemented within reasonable computing cost.

---

**Algorithm 1:** Linear-Nonlinear Explanation for any classifiers

---

**Input:** Explained model $f$, input instance $\boldsymbol{x}$
**Initialization:** $N(x_k) = 0, \ k = 1, 2, \ldots, m$
**for** $x_k \in \boldsymbol{x}$ **do**
    compute $L(x_k)$ using ( 7)
$\boldsymbol{z}_1 \leftarrow [x_*], \ x_* = \underset{x_k \in \boldsymbol{x}}{\arg\max} \, L(x_k)$
**for** $1 \leq d \leq m - 1$ **do**
    **for** $x_k \in \boldsymbol{x}$ **do**
       compute $N(x_k, \boldsymbol{z}_d)$ using (12)
       $N(x_k) \leftarrow N(x_k) + N(x_k, \boldsymbol{z}_d)\dfrac{1}{\sqrt{\pi(m-1)/2}} \exp\{-\dfrac{(d - ((m-1)/2))^2}{(m-1)/2}\}$
    $x_i = \underset{x_k \in \overline{\boldsymbol{z}_d}}{\arg\max} \, N(x_k)$
    $\boldsymbol{z}_{d+1} \leftarrow \boldsymbol{z}_d + [x_i]$
**for** $x_k \in \boldsymbol{x}$ **do**
    Compute $C(x_k)$ using (13)
**Output:** Feature's linear-nonlinear importance measured by $C(x_k)$

---

## 4 EXPERIMENTS

In this section, we conduct experiments to validate effectiveness of the proposed LNE method. Firstly, we will show the linear and nonlinear indicators are both meaningful to provide insights,

and the combined indicator provides a comprehensive understanding. Secondly, we compare LNE's explanation with other methods to see if it is consistent with human intuitions. Additionally, we do a user study to examine whether our method can hint humans of potential bias of model.

## 4.1 LINEAR AND NONLINEAR ASPECTS BOTH PROVIDE INSIGHTS

In our design, we investigate both the linear and nonlinear aspects of the model, and combine them consistently. The two aspects of feature importance are both meaningful. As shown in Figure 2, we visualize the linear and nonlinear indicators. It shows that the two indicators capture different features from the input image. By combining them into one, they complement information from each other to offer a comprehensive understanding.

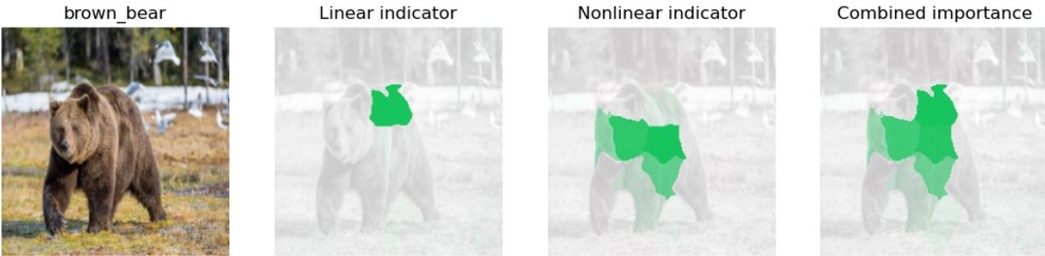

Figure 2: The linear and nonlinear indicators capture different aspects of information, and complement with each other

## 4.2 EXPLANATION COMPARISON ON WELL-TRAINED MODEL

In this part, we visually examine whether LNE's explanation for a well-trained model is consistent with human intuitions. Image classifiers trained on the ImageNet database (Deng et al., 2009) are widely used as benchmarks for visualization tasks. These models are trained on hundreds of thousands of images to learn to classify them into one thousand labels. Thus, they have a wide range of cognition, and are capable of dealing various images. For this reason, we choose one of ImageNet model, the Google Inception_v3 (Szegedy et al., 2016) as the well-trained model to explain. We assume its prediction of common images are based on human-agreed features. When explaining image classifier, super-pixel algorithm can help clustering pixels into superpixels. SLIC (Achanta et al., 2012) can generate compact, nearly uniform superpixels. Thus, we choose it to segment input images to generate interpertable features.

We compare the performance of LIME, SHAP, and proposed LNE, as shown in Figure 3. The number of superpixels is set to 50 for LIME and LNE, i.e. number of features $m = 50$. In this case, LNE calls the explained model $2 \times (50)^2 = 5000$ times. For fairness, LIME & SHAP will also sample 5000 times. In the open-source implementation, kernel SHAP will exceed maximum memory if sampled so many times, thus we replace it by partition SHAP. It can be seen that, LNE's explanation are more consistent with human intuitions.

## 4.3 USER STUDY ON WHETHER LNE HINTS POTENTIAL BIAS

To check LNE's ability on detecting biases in classifiers, we design a user study experiments. In specific, we train a binary image classifier to distinguish dogs from cats. We collect 50 images, 25 of them are dog images, the rest are cat images. However, we embed a vague watermark at random locations in half of the images. These watermarked images are **labeled** as dogs, regardless of what the image truly is. The rest images are labeled as cats. We then train a ResNet (He et al., 2016) model on these images, and in fact we get a classifier to distinguish whether the input image is watermarked or not. As a result, our biased model is able to classify all watermarked images as dogs, the others as cats.

The user study is designed as follows. First, We show users 5 watermarked dog images, 1 watermarked cat image, and 4 cat images without watermarks, and the corresponding model predictions. Then, users are asked to vote whether the image classifier could be trust or not. After this, we show

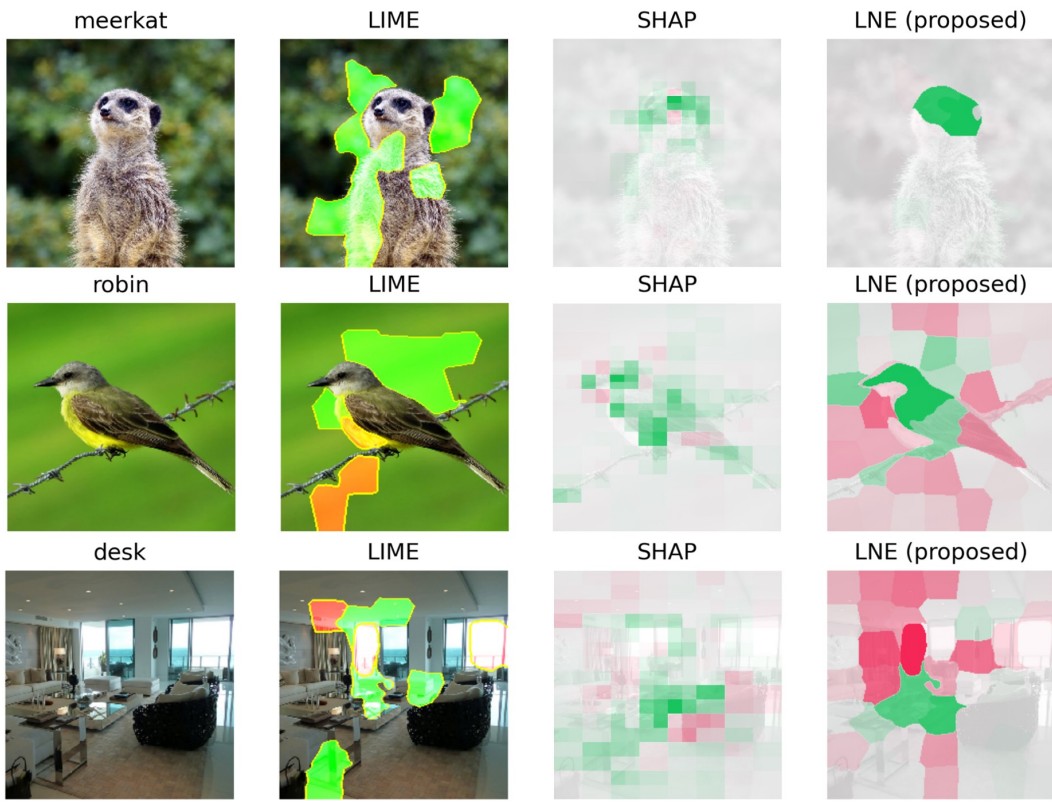

Figure 3: Performance Comparison between LIME, SHAP and LNE. Green area increases probability of the class label, red area decreases probability.

users the explanations on these images by LNE method, and they are asked to vote once again, to decide if they trust the classifier. Finally, we ask users if they notice that watermark is the true reason of model prediction. The results is shown in Table 1. As we can see, before LNE's explanations, the 10 images indicates $90\%$ accuracy of the classifier, thus the better part of users tend to trust the classifier. However, after seeing the explanations, most users are aware of potential biases, and tend to distrust it. Through the explanations, more than half users notice the watermarks. The user study shows our LNE method is capable of hinting potential bias in classifiers.

|  | Before | After |
|---|---|---|
| Trust | 14 / 19 | 2 / 19 |
| Distrust | 5 / 19 | 17 / 19 |
| Find watermark secret | 2 / 19 | 13 / 19 |

Table 1: User study on whether a biased classifier can be trusted, before and after LNE's explanations

## 5 CONCLUSION

Model-agnostic explanation methods rely on assessing feature influence to model prediction. In order to get a good explanation, feature influence must be investigated systematically.

In this paper, we propose a new explanation method that integrates independent and collaborative feature influence from linear and nonlinear aspects of the explained model. Our method approximates the linear aspect by assuming feature independence, and models the nonlinear aspect by analyzing feature interactions. The combined linear-nonlinear feature importance still preserves desirable properties for a robust explanation. Experiments validate the effectiveness of our method.

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
