# OpenReview forum: "Integrated Model Explanations by Independent and Collaborative Feature Influence via Linear-Nonlinear Perspectives."
_ICLR.cc/2024/Conference — Submitted to ICLR 2024_

### Official Review · Reviewer_5b7M · 2023-10-24

**Soundness:** 2 fair
**Presentation:** 2 fair
**Contribution:** 2 fair
**Rating:** 5
**Confidence:** 4

**Summary:**

In this paper, the authors discussed the limitations of existing works' assumption of linearity and feature independence and proposed a novel algorithm LNE, which relaxed this assumption by segregating the features into independent and collaborative groups, quantifying the feature importance of them separately and jointly. The authors experimentally demonstrated the feature importance computed through LNE aligns better with human intuition compared to the existing algorithms.

**Strengths:**

The authors identified a key limitation in existing XAI algorithms that largely assumes feature independence. To address this limitation, the authors proposed a novel algorithm to first segregate the feature importance into two chunks: marginal contribution (Linear indicator) and collaborative contribution (Nonlinear aspect). Such separation intuitively makes sense.

The authors further showed LNE has superior results on many real-life examples compared to the state-of-the-art algorithms (SHAP and LIME).

**Weaknesses:**

Though I agree with the general direction of the algorithmic design, several major parts are unclear, preventing me from further evaluation

Major:
1. Groudtruth simulation and generative model are missing. The authors proposed their algorithm to segregate the linear and nonlinear part of the contribution of each feature, but it lacks sufficient theoretical justification of why it should work (I will discuss in Questions section). In this case, it is highly suggested that the authors run their algorithm on synthetic datasets, where the linear and interactive relationship of the data is known, and test the sensitivity and specificity.

2. It is unclear how to combine the linear and nonlinear parts of the contribution in the way the authors defined. Specifically, the authors chose to use the hyperbolic tangent function to normalize the overall feature importance, which equally weights the linear contribution and the nonlinear contribution, but it is unclear why such a combination makes a good approximation of the actual feature importance. For example, can I understand the additive effect defined here as the first order taylor expansion of the black-boxed function?

Minor:
1. In section 3.2.2, the Gaussian approximation error should be O(1/sqrt(m))
2. In equation (9), vector z is not defined

**Questions:**

1. In equation (7), the marginal contribution is computed based on Theorem 1. However, theorem 1 holds under the assumption that features are independent, which doesn't match with the initial proposal the authors have: the features in the function (likely nonlinear and dependent) can be decomposed into linear and collaborative parts. Applying theorem 1 would be more proper if the authors seek to search a subset of independent features instead of decomposing a given feature's linear and nonlinear part.

2. I do not follow why equation (9) is true. How do you choose the value of xk, and why would that make sense.

3. In theory or practice, what would be the approximated feature importance accuracy compared to the ground truth when using the hyperbolic tangent function to combine the linear and nonlinear parts?

4. In algorithm 1, z_d is constructed using z_{d-1} + argmax_{x_k \in \Bar{z}_{d-1}}. Does this imply there is only one z_d vector being constructed for each d? Can you justify why this is a good approximation of the equation (10), which requires summing up all the possible z with length d, as described in the text?

---

### Official Review · Reviewer_fF3Y · 2023-10-27

**Soundness:** 1 poor
**Presentation:** 1 poor
**Contribution:** 1 poor
**Rating:** 3
**Confidence:** 3

**Summary:**

In this manuscript the authors propose LNE, a model explanation method that aims to capture the both the independent/linear aspects of individual features, as well as nonlinear/interaction effects. The authors evaluate their method qualitatively on image data and perform a user study.

Update after rebuttal period: The authors did not provide a rebuttal and (based on my original review + other reviews) I am choosing to keep my original score.

**Strengths:**

* **Significance:** The authors tackle a problem (i.e., developing methods for better understanding model behavior) that is of great importance to the ML community.
* **Novelty:** The authors' method is, to my knowledge, novel.

**Weaknesses:**

In my view the manuscript has multiple serious issues at this stage and is not ready for publication. Details below:

* **Confusion/abuse of terminology:** Throughout the manuscript the authors use "linear" interchangeably with "independent" when describing different attribution methods (and, similarly, nonlinear is used to indicate feature interactions). However, in reality these concepts are not equivalent. For example, in the original implementation of SHAP feature independence is assumed to simplify the computation of Shapley values. However, subsequent works have argued/proposed methods for calculating/approximating Shapley values without assuming feature independence (see e.g. [1,2]). Importantly, these other implementations for calculating Shapley values continue to use the _linear additive feature attribution model_ as in Definition 1 of [3]. In my view the manuscript needs a thorough pass to clarify this terminology, as currently the text does not agree with standard terminology in the feature attribution literature.
* **Missing literature/discussion of previous work:** As mentioned in my previous point, many works have attempted to address issues with the feature independence assumption of SHAP (see [4] for a review). However, none of these previous methods are discussed or compared with the authors' proposed method.
* **Experimental issues:** The authors only briefly evaluate their method qualitatively with three images (Figure 3) and with a user study, and these results are far too limited to assess the authors' method. For instance, in the comparisons between LIME/SHAP/LNE, it appears that different segmentation methods are used to construct the superpixels considered by each method. Thus, the provided results cannot be compared across methods, as it is impossible to disentangle the effect of the segmentation algorithm from the underlying method (i.e., LIME/SHAP/LNE). Moreover, as noted previously, _many_ other attribution methods have been proposed since LIME/SHAP, and thus it is not sufficient to only compare against LIME/SHAP (also, no methods _at all_ besides LNE are included in the user study). Finally, a small subset of qualitative results is not sufficient for evaluating the quality of a proposed method. Standard quantitative benchmarks (e.g. ROAR [5]) have existed for many years now, and must be used to provide more objective evaluations of interpretability methods' performance.

[1]: Aas et al., “Explaining individual predictions when features are dependent: More accurate approximations to Shapley values” (2019)
[2]: Frye et al., “Shapley-based explainability on the data manifold” (2020)
[3]: Lundberg and Lee., "A Unified Approach to Interpreting Model Predictions" (2017)
[4]: Covert, Lundberg, and Lee., "Explaining by Removing: A Unified Framework for Model Explanation" (2021)
[5]: Hooker et al., "A Benchmark for Interpretability Methods in Deep Neural Networks" (2018)

**Questions:**

* How does the authors' proposed method fit in with the previous literature attempting to address the assumption of feature independence in SHAP?
* Do the authors have additional experimental results that they could provide?
* Were any other methods considered in the user study?

---

### Official Review · Reviewer_84DP · 2023-10-30

**Soundness:** 2 fair
**Presentation:** 3 good
**Contribution:** 2 fair
**Rating:** 3
**Confidence:** 4

**Summary:**

The authors derive a feature-importance explanation measure that considers so-called linear and nonlinear relationships between the features and the prediction. _Linear_ and _nonlinear_ meaning, respectively, a one-to-one or  many-to-one relationships.
The method builds upon SHAP's idea of locally linearizing the model. The difference lies in that the proposed measures consider the difference in predictions between a set of features and its complement.

**Strengths:**

* The idea of comparing a set of features to its complement is interesting and opens interesting insights.
* The text is easy to follow despite typos and missing words.

**Weaknesses:**

* The proofs are not clear which weakens the statements and the whole proposal.
* The proposed measures both mixes terms with different expected behavior. What we get is necessarily a mashup of all which becomes not obvious to disentangle. Sadly, no analysis is provided.
* The evaluation is short and not conclusive.
* I am not a psychologist/social scientist, however, the setting of the user study does not convince me nor its results.

**Questions:**

* P1§1: In the introduction you state a question: "Do these methods faithfully adhere to the
original model and provide accurate explanations?" Do you answer it?
* P2§2: "We’d like to propose" Please avoid such abbreviations.
* P3S2.: Why do you cite Deng 2012 which introduced ImageNet when talking about MNIST?
* P4S2.3 Eq3: What is $z'$?
* P5 Def2: we have $z \subseteq y \subset x$ and $x_k \not \in z$. I understand from what follows that you don't want $x_k$ in $y$, either. Am I correct? In this case, you could write $x_k \not \in y$ which will also have it out of $z$.
* P5.S3.1.3: You describe the "meaning" of $f([x_k])$, of $f(x) - f(x \setminus [x_k])$ and of $f([x_k]) - f(x \setminus [x_k])$, but then $L$ is just the sum of these. This means their behavior is mixed and it is not clear how $L$ behaves, or what it captures. I believe you need to study the phenomenon to justify the formula. Otherwise, the discussion of the three terms taken separately is caduc.
* P6S3.2.1: What is $\mathcal{I}$? What does it measure? and why is its derivative known (Eq9)?
* P6S3.2.1: "with & without" please avoid.
* P6S3.2.1: "However, we only concern certain feature $x_k$ ’s importance within $\mathcal{I}$" I don't understand that sentence. Can you reformulate it?
* P6S3.2.2: You assume that $P(z) = 2^{-(m-1)}$, this means that $\sum_{|z|=d} P(z) = \binom{m}{d}  2^{-(m-1)}$. This is easy to compute, so why do you approximate it with Eq12?
* P7Eq13: Again you are naively combining two functions that may have very different behaviors. It is later difficult to disentangle any insight from that combined measure.
* P7Th2 Proof: Why do we have: "Let $z = \emptyset$ and $z = x\setminus [x_k ]$, then $L' (x_k ) \geq L(x_k )$.
* P7Th2 Proof: "With fixed sampling trace, $N'(x_k , z_d ) \geq N (x_k , z_d )$ is always true", this is what you need to prove. You can't just state it like that.
* S4: Why do you consider only LIME and SHAP as baselines?
* S4.2: Why such a sort of analysis, besides not necessarily clear?
* S4.3: I am not convinced that this is an unbiased experimental setting.

---

### Official Review · Reviewer_hDKN · 2023-10-31

**Soundness:** 2 fair
**Presentation:** 2 fair
**Contribution:** 1 poor
**Rating:** 3
**Confidence:** 4

**Summary:**

This work considers the topic of providing transparency for black-box machine learning models, and it proposes an approach called "linear-nonlinear explanation" (LNE) to provide a comprehensive understanding of feature influence. The method is grounded in a new notion of "feature independence" (which differs from common assumptions of statistical independence of features, or the idea of constant marginal contributions), and uses a tanh function to aggregate quantities representing the independent and collaborative importance for each feature.

**Strengths:**

It's true that most model explanation methods fit simple proxies to a model (e.g., additive functions) that sacrifice information to provide concise explanations to the user. It's useful to explore solutions that provide more information, particularly about the interactions between features. This work proposes a new perspective on how to do this.

**Weaknesses:**

The main issue is that there are many works considering feature interactions in recent years, and this paper doesn't refer to any of them. It is therefore difficult to understand the differences with this work and the potential advantages of this method. For example, see "Explaining Explanations: Axiomatic Feature Interactions for Deep Networks", "An axiomatic approach to the concept of interaction among players in cooperative games", "Hierarchical interpretations for neural network predictions", "Joint Shapley values: a measure of joint feature importance", "The Shapley Taylor interaction index", "A measure of added value in groups", and "Axiomatic characterizations of generalized values." Many of these works are based on similar game-theoretic formulations and are derived from similar axioms to the Shapley value, so it's important to know how this work differs from them.

Another issue is that this method is based on the property described in Definition 2, which the authors call "feature independence." First, it is worth remarking that this term already means something in explainable machine learning, and it's about how missing features are handled: for example, KernelSHAP (introduced by Lundberg & Lee) samples features from their marginal rather than conditional distribution, which amounts to an assumption of statistical independence. It's a confusing choice by the authors to overload this term with their definition 2. Second, their version of this property does not seem useful: the authors provide no argument that this property holds in practice for realistic model, and it does not suggest a unique additive decomposition - the authors only propose one in Theorem 1, but there are others.

Besides this, there are many issues with the work that would need to fixed before it's suitable for publication. To describe a couple:

- Theorem 1 is a small result, it only claims that these coefficients sum to the desired value. That doesn't merit being called a theorem, it would be better labeled as a proposition. It's also unclear why we should choose these coefficients, as there are other choices of coefficients that sum to the correct value, and we might prioritize coefficients that lead to a good additive proxy on average for many  $z$ values. Also, considering my point from above that definition 2 may not be a reasonable assumption for realistic models, this method overall seems to lack justification. For example, after reading this I cannot explain why we shouldn't just use the Shapley value for the linear part of the explanation.

- Regarding the discussion about SHAP in the introduction: weighted linear regression is only one of many ways to estimate Shapley values when the feature dimension is large, see "Algorithms to estimate Shapley value feature attributions" for a recent review covering many more approaches.

- Regarding the discussion of related work in the introduction: the authors draw a distinction between methods based on linear simplification and "analyzing feature influence to model prediction." I've read many of these papers and don't see the difference, there doesn't appear to be any important distinction here. The discussion seems intended to show flaws with both sides that this work can help solve. But the robustness issue discussed by Alvarez-Melis & Jaakkola is not addressed, because this flaw would still hold for the new method (see "On the Robustness of Removal-Based Feature Attributions" for why this issue often reduces to the model's Lipschitz continuity). And the flaws for the second group of methods, including the requirement of a baseline or distribution over inputs, also hold for methods in the first group, including SHAP! Overall, it's a confusing overview of the literature that fails to set up the contributions of this work.

- It's unclear whether the scales for the linear and non-linear components are actually different, or why this is the case. Both are in units of the function output, so why would they have very different scales? Additionally, if the scales are very different, it's unclear why the tanh function would mitigate this: this could result in some of the values being near the extreme values 1 or -1 while others are near 0. Overall, this seems like a random heuristic without a clear motivation. If it's truly important, you should be able to explain why and demonstrate it in your experiments, rather than leaving it for future researchers to tinker with and try to understand.

- It's unclear why the interaction values need to be calculated at all if the final result simply involves aggregating them with the linear scores. Methods like the Shapley value automatically capture interactions in their scalar scores, so don't they serve a similar purpose? This method seems like it needlessly introduces extra steps.

- Writing $f = (\mathcal{L}, \mathcal{N})$ in Section 3 is strange, it looks like a mathematical claim but it doesn't appear to mean anything. I would suggest either explaining this notation more clearly or removing it.

- The writing overall contains a lot of fluff, including words like "comprehensively" and "consistently" that don't mean anything specific. This could be cut back.

**Questions:**

Several questions were raised in the weaknesses section above.

---

### Official Review · Reviewer_9qFs · 2023-11-01

**Soundness:** 1 poor
**Presentation:** 2 fair
**Contribution:** 1 poor
**Rating:** 3
**Confidence:** 4

**Summary:**

The authors present a method (LNE) for generating explanations of predictions for black-box supervised learning models by decomposing the contribution of a particular feature to the model output into a "linear" (direct) contribution and "nonlinear" (indirect) contribution. Mathematical derivations of the method are presented. Normalization of the importances is achieved with a tanh transform, ensuring that the transformed values fall within the range (-1,1).  The method is illustrated anecdotally on a handful of images and on a small scale user trust study where 50 images with some spurious-correlation watermarks added are classified and users are asked whether they trust the decisions before and after seeing the LNE explanations.

**Strengths:**

I think the general approach of decomposing explanations into direct effects and interaction effects has some promise.

The work is original to the best of my knowledge.

The design of the user trust study is thoughtful and has some value. It just needs to be done at a larger scale and needs to compare competing methods with LNE.

**Weaknesses:**

There simply aren't enough large scale experiments in this paper to justify acceptance at ICLR. Aside from the user trust study, all I see in terms of experiments are the model explanations of LNE as compared to LIME and Shapley on just a handful of images. There's no systematic, objective evidence presented on any sort of large-N test set which could demonstrate with any rigor that LNE beats competing methods.

Although I like the concept behind the user study, 50 images and 19 users is too small-scale a study to carry much weight  at a conference like ICLR.

The writing is also not particularly clear. I was able to understand the basic logic behind the paper, but there are lots of awkward phrasings here.

I encourage the authors to continue this line of research, but it's not ready for ICLR.

Some typo-level suggested edits:

Abstract:
to model prediction-> to model predictions
Is independent influence necessarily linear or “generalized linear” ?
Can hint humans?
P2 should reflect on simplified model -> should be reflected in the simplified model
P2 In specific -> Specifically
P2 We’d like to propose an method -> We’d like to propose a method
P4 discrepency-> discrepancy
P4 we focuses -> we focus
P6 we only concern-> we are only concerned with

**Questions:**

Do you have larger scale (large N), quantitative evidence for the superiority of LNE? Showing a few images of explanations is not enough for a conference paper.

Did you test LIME or Shapley in the 50 image user study?

The tanh transformation is an odd choice. Why not just standardize (subtract mean, divide by std dev) ?

---

### Meta-Review · Area_Chair_gSHR · 2023-12-06

**Metareview:**

The paper proposes Linear-Nonlinear Explanation (LNE) method, a novel approach to capturing both independent and collaborative influences on model predictions.

pros:
- The paper addresses a crucial problem in ML, focusing on improving model interpretability, which is of great importance to the community.

cons:
- lacks quantitative benchmarks, relies on small datasets, and encounters challenges in comparing results.
- lacks a clear and convincing explanation of the use of the hyperbolic tangent function for combining linear and nonlinear contributions and approximating actual feature importance.
- lacks theoretical justification or groundtruth simulation and analysis of algorithm sensitivity and specificity

**Justification For Why Not Higher Score:**

lacks empirical validation and convincing explanation/justification of algorithms

**Justification For Why Not Lower Score:**

NA

---

### Decision · Program_Chairs · 2024-01-16

Reject